# The Flavonoids and Monoterpenes from *Citrus unshiu* Peel Contained in Ninjinyoeito Synergistically Activate Orexin 1 Receptor: A Possible Mechanism of the Orexigenic Effects of Ninjinyoeito

**DOI:** 10.3390/biom15040533

**Published:** 2025-04-05

**Authors:** Kaori Ohshima, Kanako Miyano, Miki Nonaka, Sayaka Aiso, Mao Fukuda, Saho Furuya, Hideaki Fujii, Yasuhito Uezono

**Affiliations:** 1Department of Pain Control Research, The Jikei University School of Medicine, Tokyo 105-8461, Japan; kohshima@jikei.ac.jp (K.O.); k.miyano.fy@juntendo.ac.jp (K.M.); minonaka@jikei.ac.jp (M.N.); pp17001@st.kitasato-u.ac.jp (S.A.); pp16189@st.kitasato-u.ac.jp (M.F.); pp15207@st.kitasato-u.ac.jp (S.F.); 2Department of Pathology, Immunology, and Microbiology, Graduate School of Medicine, The University of Tokyo, Tokyo 113-0033, Japan; 3Laboratory of Pharmacotherapeutics, Faculty of Pharmacy, Juntendo University, Chiba 279-0013, Japan; 4Center for Neuroscience of Pain, The Jikei University School of Medicine, Tokyo 105-8461, Japan; 5Laboratory of Medicinal Chemistry, School of Pharmacy, Kitasato University, Tokyo 108-8641, Japan; fujiih@pharm.kitasato-u.ac.jp; 6Pharmacological Department of Herbal Medicine, Graduate School of Medical and Dental Sciences, Kagoshima University, Kagoshima 890-8544, Japan; 7Department of Comprehensive Oncology, Graduate School of Biomedical Sciences, Nagasaki University, Nagasaki 852-8523, Japan

**Keywords:** orexin 1 receptor, Ninjinyoeito, *Citrus unshiu* peel, flavonoid, monoterpene

## Abstract

Cancer cachexia, often observed in patients with advanced-stage cancer, is characterized by the loss of body weight and appetite. The Japanese herbal medicine Ninjinyoeito (NYT), which is composed of 12 crude herbal components, has been used as a therapeutic in Japan to improve anorexia and fatigue, which are commonly observed in cancer patients with cachexia. We have previously reported that *Citrus unshiu* peel (CUP) contained in NYT can enhance food intake by activating the orexin 1 receptor (OX1R). Using the CellKey™ system, which offers detection of OXR activity in intracellular impedance changes, NYT and CUP were found to activate OX1R, which in turn was inhibited by SB-674042, a selective OX1R antagonist. Among the flavonoids contained in CUP, nobiletin and hesperidin, but not naringin, activated OX1R. Furthermore, some monoterpenes contained in CUP, including limonene and linalool, but not terpineol, activated OX1R. In addition, nobiletin and limonene synergistically activated OX1R when added simultaneously. However, neither NYT nor CUP induced OX2R activity. The results collectively suggested that the CUP contained in NYT activates OX1R, but not OX2R, and that flavonoids and monoterpenes in CUP can synergistically activate OX1R. These findings could provide evidence supporting the therapeutic potential of NYT in cancer patients with cachexia.

## 1. Introduction

Cancer cachexia, a syndrome characterized by progressive weight loss and appetite reduction, affects approximately 80% of patients with advanced-stage cancer and accounts for at least 20% of cancer-related deaths [1,2,3]. This syndrome both compromises the quality of life (QOL) and diminishes the effectiveness of chemotherapy [4,5]. The condition is reported to arise from intricate interactions among multiple mediators in the hypothalamus, such as hormones (e.g., leptin, ghrelin, α-melanocyte-stimulating hormone, and melanin-concentrating hormone), and neuropeptides (e.g., neuropeptide Y, agouti-related protein, and orexin), which are involved in appetite regulation [6]. However, the precise mechanisms underlying this syndrome remain unclear, and effective therapeutic strategies have yet to be established. Currently available treatments for cancer cachexia remain inadequate [3].

In Japan, traditional herbal medicines, namely kampo medicines, have been prescribed and approved for clinical use by Japan’s Ministry of Health, Labour and Welfare [7]. Ninjinyoeito (NYT), one of these Japanese kampo medicines, has multifunctional beneficial activities and it has been used to improve recovery from diseases or untoward symptoms including fatigue, anorexia, and neuropathies often seen in cancer patients with cachexia [8,9,10,11]. NYT consists of 12 crude herbal ingredients, each of which has been reported to exhibit various bioactive properties, including anti-inflammatory, anti-tumor, anti-oxidative, and neuroprotective effects [10] (Table 1).

Owing to these properties, NYT improves appetite in an animal model of cancer cachexia and in patients with cancer [11,17,43,44]. We had previously reported that *Citrus unshiu* peel (CUP), one of the components of NYT, activates the orexin 1 receptor (OX1R) [11], the receptor involved in appetite promotion [10,11]. Further, we also reported that NYT did not relate to the signaling pathways caused by NPY1, NYP5, and ghrelin receptors [11]. Accordingly, we proposed CUP in NYT could activate orexigenic OX1R to consequently improve loss of appetite in cancer patients with cachexia [11] (Table 1). In addition to eliciting OX1R activity, CUP has been reported to improve the symptoms of cancer cachexic patients via a variety of mechanisms including improvement in muscle atrophy, adipose tissue atrophy, and chemotherapy-induced atrophy, as well as via the regulation of cytokine levels causing such atrophic phenomena observed in cancer cachexia [45] (Table 2).

CUP contains several functional compounds such as flavonoids, monoterpenes, and many other low molecular compounds known to have beneficial biological effects [32]. At present, however, the mechanisms by which CUP activates OX1R and the active components contained in CUP remain to be clarified.

The present study thus aimed to examine the effects of various ingredients in CUP on the activity of OX1R, in addition to another orexin-binding receptor, OX2R. We found that the CUP contained in NYT can selectively activate OX1R but not OX2R, and some flavonoids and monoterpenes in CUP can synergistically activate OX1R.

## 2. Materials and Methods

### 2.1. Chemicals and Reagents

In this study, reagents and media used were as follows: bovine serum albumin (BSA) and poly-D-lysine (Sigma-Aldrich, St. Louis, MO, USA); geneticin and fetal bovine serum (FBS) (Gibco, Carlsbad, CA, USA); penicillin/streptomycin and 4-(2-hydroxyethyl)-1-piperazineethanesulfonic acid (HEPES) (Nacalai Tesque, Kyoto, Japan); and Dulbecco’s modified Eagle’s medium (DMEM) (FUJIFILM Wako Pure Chemical Corporation, Osaka, Japan). *Citrus unshiu* peel (lot no. T160528) and NYT powder extract (lot no. 15112017) were gifted by Kracie Pharma, Ltd. (Tokyo, Japan). The NYT powder extract comprised 12 medicinal herbs in the following proportions: *Rehmannia* root (12.9%), *Japanese angelica* root (12.9%), *Atractylodes rhizome* (12.9%), *Poria sclerotium* (12.9%), *Ginseng* (9.7%), *Cinnamon* bark (8.1%), *Citrus unshiu* peel (6.5%), *Polygala* root (6.5%), *Peony* root (6.5%), *Astragalus* root (4.8%), *Glycyrrhiza* (3.2%), and *Schisandra* fruit (3.2%). Flavonoids including nobiletin, hesperidin, and naringin, as well as monoterpenes such as limonene, linalool, and terpineol, which are ingredients in CUP, were sourced from FUJIFILM Wako Pure Chemical Corporation, Japan. The dried NYT powder extracts and CUP as well as the flavonoids and monoterpenes were diluted in sterile water at a concentration of 100 mg/mL (NYT and CUP) or 10 mM (flavonoids and monoterpenes). We made these solutions 100-fold dilutions using Hanks’ balanced salt solution (composition: 5.4 mM KCl, 1.3 mM CaCl_2_·2H_2_O, 0.81 mM MgSO_4_, 4.2 mM NaHCO_3_, 0.44 mM KH_2_PO_4_, 0.34 mM Na_2_HPO_4_, 136.9 mM NaCl, and 5.6 mM D-glucose) supplemented with 20 mM HEPES and 0.1% BSA, followed by filtration with a 0.2-μm membrane (KURABO Industry Ltd., Osaka, Japan). The resulting solutions were applied to cells at the concentrations specified in the figure legends. Other reagents were sourced from commercial suppliers with the highest available purity.

### 2.2. Establishment of Stable Cell Lines

Human OX1R and OX2R clones (GenBank accession numbers AB463762 and AB463763, respectively) were obtained from the Kazusa DNA Research Institute, (Chiba, Japan) and amplified following guidelines from manufacturers. HEK293 cells (American Type Culture Collection, Manassas, VA, USA), that stably express OX1R were generated with plasmid transfection using ScreenFect^TM^ (FUJIFILM Wako Pure Chemical Corporation, Osaka, Japan), and selected according to the CellKey^TM^-based OX1R activity assay. OX2R-expressing cells were developed by the Basis for Supporting Innovative Drug Discovery and Life Science Research (BINDS), and their functionality was validated using the CellKey^TM^ assay (MDS Sciex, Concord, ON, Canada). Ethical approval for all experimental procedures was obtained (approval no. B85M1-13 from the National Cancer Center Research Institute).

### 2.3. Cell Culture

Cells used in the present study were maintained in a humidified environment containing 95% air and 5% CO_2_ at 37 °C. HEK293 cells expressing OX1R and OX2R were cultured in DMEM with 10% FBS, 100 U/mL penicillin, 100 mg/mL streptomycin, and 800 μg/mL geneticin supplementation.

### 2.4. Measurement of OX1R and OX2R Activity, and GPCR Families Using the CellKey^TM^ System

The evaluation of OX1R and OX2R activities using the CellKey™ assay was conducted following previously described methods [11,47,48]. There are various GPCR assay systems, including measurements of intracellular cAMP levels, Ca^2+^ concentrations and GTPγS activity [49]. Additionally, cellular dielectric spectroscopy (CDS) is an advanced label-free, real-time, cell-based assay, particularly suited for GPCR activation assays [49,50]. The CellKey™ system, a type of CDS, utilizes impedance-based biosensors and provides a cost-effective solution for GPCR assays, including those targeting Gs-, Gi/o-, and Gq-coupled receptors [49,50,51]. In this study, HEK293 cells expressing OX1R or OX2R were seeded at a density of 6.0 × 10^4^ cells per well in the 96-well microplates for the CellKey^TM^ assay. Following a 24 h incubation at 37 °C, the cells were rinsed with Hanks’ balanced salt solution supplemented with 20 mM HEPES and 0.1% BSA, then equilibrated for 30 min in the assay buffer before the assay. During the assay, the CellKey^TM^ apparatus applied small voltage pulses every 10 s to measure impedance of cells. Initially, a baseline impedance was recorded for 5 min before drug administration, after which cellular impedance (ΔZ) changes were monitored for 25 min. The impedance change rate was determined as the difference between the minimum and maximum impedance values following drug treatment, in accordance with previous reports [47,49].

### 2.5. Statistical Assessment

Data are expressed as means ± S.E.M. For statistical comparisons, a one-way analysis of variance (ANOVA) was carried out, followed by Bonferroni’s multiple comparison test with GraphPad Prism 8 (GraphPad Software, San Diego, CA, USA). A significance level of *p* < 0.05 was used to determine statistical significance.

## 3. Results

### 3.1. NYT and CUP Activated OX1R

Our previous study demonstrated that both NYT and CUP activate OX1R [11]. In this study, we further evaluated their effects on OX1R activation using the CellKey™ system. As shown in Figure 1A, treatment with NYT (100 µg/mL) and CUP (20 µg/mL) led to an increase in the ΔZ value of OX1R-expressing cells, consistent with our prior findings [11]. Moreover, the OX1R antagonist SB-674042 (SB, 10^−6^ or 10^−5^ M) significantly inhibited the CUP-induced increase in ΔZ in a dose-dependent manner, confirming that the effect was mediated via activation of OX1R (Figure 1B).

### 3.2. Flavonoids Contained in CUP of NYT Activated OX1R

CUP contains several flavonoids and monoterpenes [32]. We chose the major flavonoids present in CUP, namely nobiletin, hesperidin, and naringin, for the study. All three flavonoids have been reported to possess biological activity, including depression suppression, anti-cancer effects, beneficial effects in Alzheimer’s disease, and overall antioxidant activity [32,52]. As shown in Figure 2, nobiletin elicited OX1R activity (Figure 2A) in a dose-dependent manner. Hesperidin also exhibited OX1R-activating effects, though weaker than nobiletin (Figure 2). In contrast, naringin did not activate OX1R (Figure 2C).

### 3.3. Some Monoterpenes Activated OX1R

Next, we examined the effects of monoterpenes, which are the main components of CUP [32]. We selected three monoterpenes and analyzed their effects on OX1R activity. As shown in Figure 3, limonene and linalool, but not terpineol, activated OX1R in a dose-dependent manner (Figure 3). To further assess the synergistic effects of ingredients in CUP on OX1R activity, limonene and nobiletin were checked and synergistic OX1R activating effects were found in OX1R-expressing cells (Figure 4). Furthermore, the simultaneous activation of OX1R by both ingredients was inhibited by the OX1R antagonist SB-674042 in a dose-dependent manner (Figure 4).

### 3.4. Neither NYT nor CUP Activated OX2R

We next examined whether NYT influences OX2R, another orexin-binding receptor that functions alongside OX1R. In OX2R-expressing cells, orexin activated OX2R in a dose-dependent manner (Figure 5A). This activation was almost completely suppressed by the OX2R antagonist TCS-OX2-29 (Figure 5B), confirming the specificity of the response.

As shown in Figure 5C, NYT induced a slight activation of OX2R at high concentrations (100 µg/mL); however, this increase was not inhibited by TCS-OX2-29 at concentrations that fully blocked orexin-induced OX2R activation (Figure 5D). Furthermore, CUP at a concentration that induces OX1R activity (Figure 1A) [11] failed to activate OX2R, and the OX2R antagonist had no effect on CUP-induced OX2R activity (Figure 5E).

## 4. Discussion

We had previously reported that CUP, one of the 12 components of NYT, activates orexigenic receptor OX1R [11]. The orexigenic peptide orexin is known to accelerate appetite [53,54,55,56]. The present study showed that some flavonoids (nobiletin and hesperidin) and monoterpenes (limonene and linalool), which are the main components of CUP [32], activate OX1R, which can be inhibited by the OX1R antagonist SB-674042. Further, synergistic effects of nobiletin and limonene on OX1R activity were observed. These results suggested that NYT promotes appetite through CUP-induced OX1R activation via the synergistic appetite-promoting activities among flavonoids and monoterpenes contained in CUP.

Both NYT and CUP have been reported to alleviate symptoms of cancer cachexia, including appetite loss [44,45,46]. In aqueous extracts of CUP, the predominant flavonoids include hesperidin, nobiletin, tangeretin, heptamethoxyflavone, naringin, and synephrine [57,58]. Pharmacokinetic studies have demonstrated that these flavonoids are absorbed into the bloodstream following oral administration in humans [59,60]. Moreover, several reports indicate that nobiletin and polymethoxyflavones are capable of crossing the blood–brain barrier (BBB) and reaching the brain in animal models [61,62]. Further, Shimizu et al. documented that nobiletin had high permeability in the BBB due to its high lipid solubility [63]. These results suggest that CUP-derived flavonoids may across the BBB and activate OX1R in neuronal tissues.

In addition to flavonoids, CUP contains several monoterpenes, such as limonene, linalool, and terpineol [32]. These monoterpenes have been reported to elicit biological activity, including the suppression of depression, anti-cancer effects, beneficial effects in Alzheimer’s disease, and overall antioxidant activity [32,52]. In particular, limonene has been reported to exert antianxiety activity through adenosine A2A receptors, based on the suppression of limonene-induced effects by an A2A selective antagonist [64]. Our results suggested that certain monoterpenes could activate OX1R; limonene had much stronger activity than linalool, and terpineol showed almost no effect in the present study.

Saini et al. highlighted in their review that the synergistic effects on bioavailability and bioactivity among various bioactive compounds in citrus fruits require further elucidation [32]. Nobiletin exerts synergistic anti-inflammatory effects with docosahexaenoic acid (DHA) and sulforaphane [65,66]. Currently, the synergistic mechanism of flavonoids and monoterpenes on OX1R activity remains uncertain. Further studies would be required to confirm this hypothesis.

Beyond its well-documented role in appetite regulation, orexin has been implicated in novel therapeutic strategies for inflammatory and neurodegenerative disorders, such as Alzheimer’s disease, multiple sclerosis, inflammatory bowel disease, and various cancers, through its combined anti-inflammatory and neuroprotective effects [67]. Further, the orexin receptor signaling system could be a target for the development of novel therapeutics for neuropsychiatric and neuro-degenerative diseases, based on the multifunctional properties of orexin regarding a wide range of neuronal activity [68]. From our present study, it is speculated that the multifunctional properties of NYT which appeared in the clinical field may be at least in part due to the increased activity of OX1R caused by ingredients contained in CUP.

In our study, CUP and NYT activated OX1R [11] but not OX2R. Both the protein and mRNA levels of OX1R and OX2R in the central nervous system was previously reported to be distributed throughout the rat brain in an overlapping pattern [69]. Although both OX1R and OX2R are thought to be involved in orexigenic signaling pathways in the hypothalamus, the effects of CUP and NYT on OX1R and OX2R activity are different. Rayat Sanati et al. reported the distinct functions of OX1R and OX2R in the rat hippocampal dentate gyrus, where maintenance of a morphine reward is regulated by OXRs. They showed that a blockade of OX1R shortened the extinction latency of a morphine-induced conditional place reference, while blockade of OX2R did not, indicating that OX1R but not OX2R facilitates the morphine-induced reward [70]. In contrast, in the same area of the hippocampal dentate gyrus, stress-induced antinociceptive responses in an acute pain model were mediated by both OX1R and OX2R in the same manner [71]. The results suggested that OX1R and OX2R play important roles in similar or different ways in several neuronal functions.

Limitations of this study are as follows: although we found ingredients contained in CUP activated OX1R but not OX2R, mechanisms of their different action on OX1R and OX2R were not clarified. In addition, the synergistic mechanisms of nobiletin and limonene on OX1R activity are uncertain at present. Also, it is not known why only some of the flavonoids and monoterpenes, such as nobiletin and limonene, had OX1R-activating properties. Further concise experiments on CUP-mediated improvement of anorexia via OX1R activation are required.

The orexigenic peptide ghrelin is known to improve the symptoms of cancer cachexia [72], in addition to increasing appetite in patients with cancer [73]. We had previously reported that the Japanese herbal medicine Rikkunshito (RKT) is a ghrelin receptor signaling enhancer and revealed the mechanisms by which RKT ameliorates anorexia in a model of cancer cachexia [74,75,76]. Atractylodin, an ingredient in the Atractylodes rhizome contained in RKT, was found to enhance ghrelin receptor-mediated signaling [72]. Similarly, the present study suggested that some flavonoids and monoterpenes in the CUP contained in NYT could enhance appetite through the activation of OX1R. RKT and NYT have been reported to be effective in improving the symptoms of cancer cachexia [10,11,72,73]. Taken together, these findings suggest that specific Japanese herbal medicines hold promise as therapeutic interventions for cancer cachexia. Further investigation into the mechanisms by which these herbal medicines improve cancer cachexia symptoms could provide valuable insights for future research and clinical applications.

## 5. Conclusions

We showed that the CUP contained in NYT activates OX1R, but not OX2R, and that nobiletin and limonene, which are the main components of CUP, synergistically activate OX1R. The study provided further evidence supporting the potential of CUP contained in NYT for improving cancer patients with cachexia and anorexia.

## Figures and Tables

**Figure 1 biomolecules-15-00533-f001:**
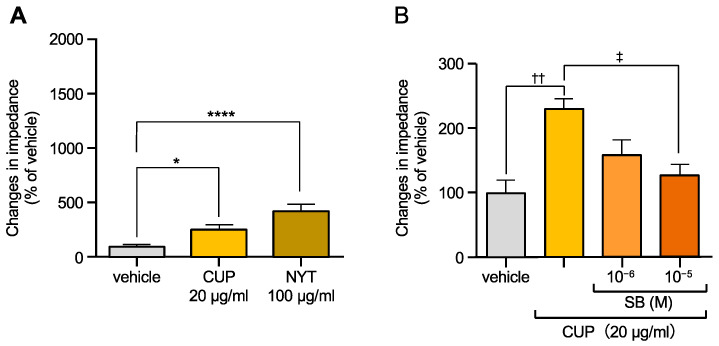
Effects of Ninjinyoeito (NYT) and *Citrus unshiu* peel (CUP) on impedance changes in cells expressing orexin type 1 receptors (OX1R) ((**A**), *n* = 9–12), and inhibition of CUP-induced OX1R activities by SB-674042 (SB) ((**B**), *n* = 9–26). Data are represented as mean ± SEM. (**A**) * *p* < 0.05 vs. vehicle, **** *p* < 0.0001 vs. vehicle (**B**) ^††^ *p* < 0.01 vs. veh;cle, ^‡^ *p* < 0.05 vs. CUP alone, Bonferroni’s multiple comparison test. The gray column shows the vehicle, the yellow column shows CUP and the brown one shows NYT. Different orange colors show CUP with different concentrations of SB-674042.

**Figure 2 biomolecules-15-00533-f002:**
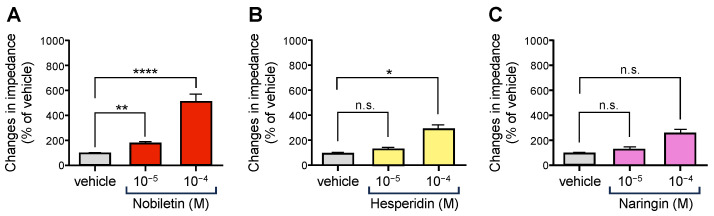
Effects of several flavonoids contained in *Citrus unshiu* peel (nobiletin, hesperidin, and naringin) on impedance changes in cells expressing OX1R (*n* = 7). Data are presented as mean ± SEM; (**A**) ** *p* < 0.01 vs. vehicle; **** *p* < 0.0001 vs. vehicle (**B**) * *p* < 0.05 vs. vehicle; n.s., not significant (**C**) n.s., not significant. One-way ANOVA followed by Bonferroni’s multiple comparisons test. The gray column shows the vehicle, the magenta column shows nobiletin, the yellow shows hesperidin, and the pink shows naringin.

**Figure 3 biomolecules-15-00533-f003:**
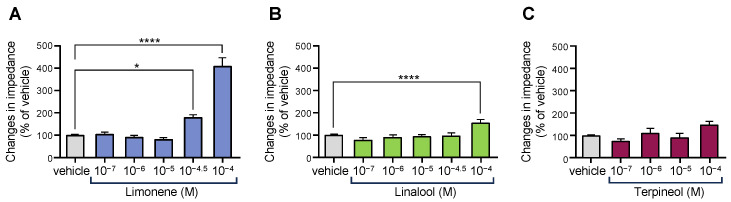
Effects of several monoterpenes contained in *Citrus unshiu* peel [limonene (**A**), linalool (**B**), and terpineol (**C**)] on impedance changes in cells expressing OX1R (*n* = 9–26). The data are represented as mean ± SEM; (A) * *p* < 0.05 vs. vehicle; **** *p* < 0.0001 vs. vehicle (**B**) **** *p* < 0.0001 vs. vehicle. Bonferroni’s multiple comparison test. The gray column shows the vehicle, the blue column shows limonene, the green shows linalool, and the dark red shows terpineol.

**Figure 4 biomolecules-15-00533-f004:**
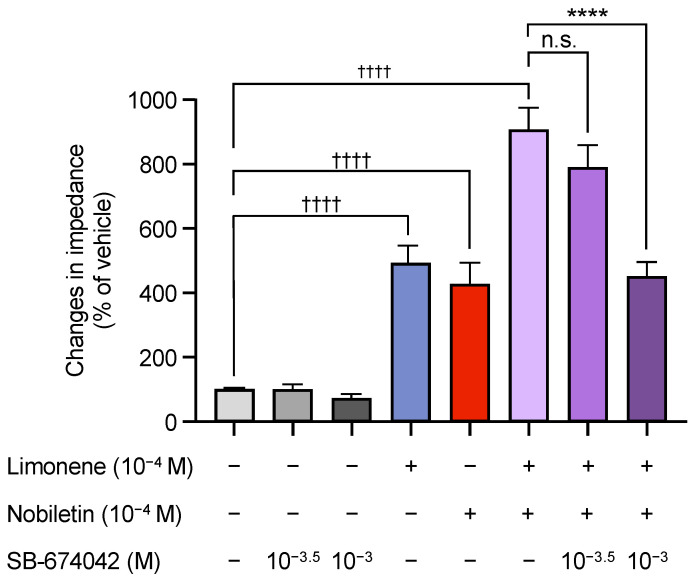
Synergic effects of limonene and nobiletin in *Citrus unshiu* peel on impedance changes in cells expressing OX1R (*n* = 7). Data are presented as mean ± SEM; ^††††^ *p* < 0.0001 vs. vehicle. **** *p* < 0.0001 vs. limonene alone; n.s., not significant. One-way ANOVA followed by Bonferroni’s multiple comparisons test. The gray column shows the vehicle, the dark gray shows SB-674042 alone, the blue shows limonene, the red shows nobiletin, the purple shows limonene + nobiletin, and the dark purple shows limonene + nobiletin with different concentrations of SB-674042.

**Figure 5 biomolecules-15-00533-f005:**
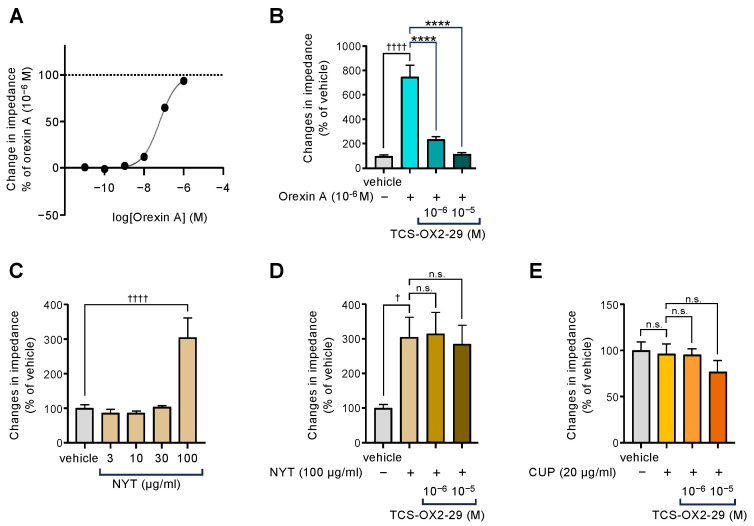
Effects of orexin A (**A**) Ninjinyoeito (NYT), and (**C**) *Citrus unshiu* peel (CUP) on (**E**) impedance changes in cells expressing orexin type 2 receptors (OX2R); and inhibition of orexin A- (**B**), NYT- (**D**), and CUP-induced OX2R activities by TCS-OX2-29 (**E**). Data are presented as mean ± SEM (*n* = 9 each); (**B**) ^††††^ *p* < 0.0001 vs. vehicle; **** *p* < 0.0001 vs. orexin A alone (**C**) ^††††^
*p* < 0.0001 vs. vehicle (**D**) ^†^
*p* < 0.05 vs. vehicle, n.s., not significant. (**E**) n.s., not significant. One-way ANOVA followed by Bonferroni’s multiple comparisons test. The gray columns show the vehicle. The blue columns show the results by orexin A, the brown columns show the results by NYT, and the orange columns show the results by CUP.

**Table 1 biomolecules-15-00533-t001:** Physiological function of ingredients from each component composing NYT.

	Herbal Components	Formula in NYT	Main Ingredients	Functions or Sites of Action	References
1	Rehmannia Root	4.0 g	catalpol	• Antineurodegenerative• Anti-ischemia-induced oligodentrocyte damage by Na^+^/Ca^2+^ exchanger 3	[12,13,14]
2	Japanese Angelica Root	4.0 g	ligustilide	• Anti-inflammatory	[15,16]
3	Atractylodes Rhizome	4.0 g	atractylenolide	• Improve symptom of cancer patients• Anti-inflammatory	[17,18,19]
4	Poria Sclerotium	4.0 g	pachymic acid	• Antitumor• Inhibition of enzymes from active acyl ghrelin to inactive des-acyl ghrelin	[20,21,22]
5	Ginseng	3.0 g	ginsenoside	• Antitumor• Anti-inflammatory• Antioxidative	[9,23,24,25]
6	Cinnamon Bark	2.5 g	cinnamaldehyde	• Anti-inflammatory• Antioxidative• Antitumor• Neuroprotective	[26,27,28]
7	*Citrus unshiu* Peel 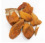 *	2.0 g	flavonoidsmonoterpenes	• Orexin 1 receptor activation• Neuroprotective• Antioxidant• Anti-inflammatory	[11,29,30,31,32]
8	Polygala Root	2.0 g	tenuigenin	• Neuroprotective• Anti-inflammatory	[33,34]
9	Peony Root	2.0 g	paeoniflorin	• Pain relief• Ca^2+^ channel inhibition	[35,36]
10	Astragalus Root	1.5 g	astragaloside,isoastragaloside	• Elevation of adiponectin production• Antitumor	[37,38]
11	Glycyrrhiza	1.0 g	glycyrrhizinglycycoumarin	• Ant-inflammatory• Antioxidative• Neuroprotective• Keep ghrelin levels as pachymic acid	[20,39,40]
12	Schisandra Fruit	1.0 g	schizandrin	• Ant-inflammatory• Enhancement of skeletal muscle endurance	[41,42]

* Picture of *Citrus unshiu* peel contained in Ninjinyoeito.

**Table 2 biomolecules-15-00533-t002:** Proposed mechanism of *Citrus unshiu* peel for improvement of symptoms of cancer cachexia.

	Sites of Function	Experimental Modes	Mechanisms	References
1	Improve muscle atrophy	Mice cachexia model	Decreased levels of TNF-α, IL-6, IL-1β	[46]
2	Improve adipose tissue atrophy	Mice cachexia model	Decreased levels of TNF-α, IL-6, IL-1β	[46]
3	Ameliorate chemotherapy-induced atrophy	Mice cachexia model	Decreased levels of IL-6, TNF-α, IL-1β, malondialdehyde-thiobarbituric acid (MDA)	[44]

## Data Availability

All datasets generated for this study have been included in this article.

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
