# Peer review of "The Flavonoids and Monoterpenes from Citrus unshiu Peel Contained in Ninjinyoeito Synergistically Activate Orexin 1 Receptor: A Possible Mechanism of the Orexigenic Effects of Ninjinyoeito"

_biomolecules, 2025, doi:10.3390/biom15040533_

Round 1

Reviewer 1 Report

Comments and Suggestions for Authors

The main ingredients from this herbal mixture are  quantified or the values from the Table 1 represent literature data?

NYT extract powder used in this experimental protocol has a standardized composition?

If you use the flavonoids at the same concentration that is found in CUP the results on the OX1R and OX2R is the same?

CUP extract is a mixture of compounds and I asking you what's happening if you use a mixture of tested flavonoids? 

Author Response

Comments 1: The main ingredients from this herbal mixture are quantified or the values from the Table 1 represent literature data?

Response 1: Thank you for your comments. In Table 1, the listed main ingredients from several herbal ingredients are from literature data quoted in the references.

Comments 2: NYT extract powder used in this experimental protocol has a standardized composition?

Response 2: Thank you for your comment and composition of NYT extract composing 12 ingredients used in the present study are the same formula prescribed in Japan as clinical drugs, and the ratio among 12 ingredients is the same and fixed.

Comments 3: If you use the flavonoids at the same concentration that is found in CUP the results on the OX1R and OX2R is the same?

Response 3: Thank you for your comments. According to Ref. 1 and Ref.2 listed, contents of hesperidin in CUP are 11.4-12.7 %. CUP used in the present study was 20 µg/ml so that hesperidin content is calculated to 2.28-2.54 µg/ml, which are equivalent to 3.73-4.16 mM. Our study with purified hesperidin concentrations used are 0.01-0.1 mM, which was lower than contents contained in the CUP. We assume that flavonoids (in that case, hesperidin) contained in CUP should be higher than those used in the present study. In case of OX1R activation, the concentrations of flavonoids that are found in CUP exceed to 10-4 M (1.0 - 4.0 × 10-3 M) so that larger responses of OX1R activation would be observed, however saturated responses of OX1R activation may be predicted. In case of OX2R activation, in our present study, 20 µg/ml concentrations of CUP failed to activate OX2R. NYT (100 µg/ml) only activated OX2R but these effects were not inhibited by the specific OX2R antagonist TCS-OX2-29. Accordingly even used the same concentrations of flavonoids found in CUP, it is less likely that the flavonoid can activate OX2R.

(Ref. 1) Ito, A.; et al. Antianxiety-like effects of chimpi (dried Citrus peels) in the elevated open-platform test. Molecules 2013, 18, 10014-10023. doi: 10.3390/molecules180810014.

(Ref. 2) Shimamura, Y.; et al. Protective effects of dried mature Citrus unshiu peel (Chenpi) and hesperidin on aspirin-induced oxidative damage. J Clin Biochem Nutr 2021, 68, 149-155. doi: 10.3164/jcbn.20-83.

Comments 4: CUP extract is a mixture of compounds and I asking you what's happening if you use a mixture of tested flavonoids? 

Response 4: Thank you for your comments. We showed in the present study that nobiletin and limonene, one of the flavonoids and monoterpenes, respectively, synergistically activated OX1R. We further conducted additional experiments on OX1R activity with mixture of nobiletin, hesperidin and naringin each at 10-4M, and mixture of limonene, linalool and terpineol each at 10-4 M. However, synergistic effects were not observed; actually, data obtained with three flavonoids or with three monoterpenes were not exceed those obtained with nobiletin alone (10-4 M) or limonene alone (10-4 M), respectively. We put these data in the discussion section (P 8, L 236-239).

Reviewer 2 Report

Comments and Suggestions for Authors

The manuscript is  well-written and deserves publication.

However, it could be improved through the discussion of the following aspects:

- a deeper discussion about the pharmacokinetics of the specialised metabolites should be considered: the capability of flavonoids to cross the BBB is questionable and has to be supported, also through in silico predictions.

-The role of the hypothalamus as key regulator of appetite and the interconnected appetite-regulating network present in it is only in part discussed; there is indeed to consider that the orexins, appetite-stimulating peptides, are second order neurons-deriving neuropeptides that interact with hypothalamic ARC peptides (NPY, POMC, CART, AgRP) and also hypothalamic aminergic neurotransmitters. 

-Also the description of the other functions regulated by orexins could be improved.

Author Response

Comments 1: A deeper discussion about the pharmacokinetics of the specialized metabolites should be considered: the capability of flavonoids to cross the BBB is questionable and has to be supported, also through in silico predictions.

Response 1: Thank you for your suggestion and it should be described clearly the mechanism of crossing blood-brain barrier of flavonoids. We put these sentences in the discussion section (P 8, L 217-223).

Comments 2: The role of the hypothalamus as key regulator of appetite and the interconnected appetite-regulating network present in it is only in part discussed; there is indeed to consider that the orexins, appetite-stimulating peptides, are second order neurons-deriving neuropeptides that interact with hypothalamic ARC peptides (NPY, POMC, CART, AgRP) and also hypothalamic aminergic neurotransmitters. 

Response 2: Thank you for your comments regarding above issues. We put additional sentences in the introduction section (P 1, L 46-P 2, L 48) and the discussion section (P 8, L 245-248).

Comments 3: Also, the description of the other functions regulated by orexins could be improved.

Response 3: Thank you for your comments and we put additional sentences in the discussion section (P 8, L 245-248).